# The Association of Gene Variants in the Vitamin D Metabolic Pathway and Its Interaction with Vitamin D on Gestational Diabetes Mellitus: A Prospective Cohort Study

**DOI:** 10.3390/nu13124220

**Published:** 2021-11-24

**Authors:** Minjia Mo, Bule Shao, Xing Xin, Wenliang Luo, Shuting Si, Wen Jiang, Shuojia Wang, Yu Shen, Jinhua Wu, Yunxian Yu

**Affiliations:** 1Department of Public Health, and Department of Anesthesiology, Second Affiliated Hospital of Zhejiang University School of Medicine, Hangzhou 310058, China; minjiamo@zju.edu.cn (M.M.); shaobl@163.com (B.S.); 21818488@zju.edu.cn (X.X.); 21918155@zju.edu.cn (W.L.); 21818499@zju.edu.cn (S.S.); wangsj2015@163.com (S.W.); shenyu_yoner@126.com (Y.S.); 2Department of Epidemiology & Health Statistics, School of Public Health, School of Medicine, Zhejiang University, Hangzhou 310027, China; 3Department of Gastroenterology, Sir Run Run Shaw Hospital, School of Medicine, Zhejiang University, Hangzhou 310058, China; 4Zhoushan Maternal and Child Care Hospital, Zhoushan 316004, China; 21618436@zju.edu.cn (W.J.); 2012302170059@whu.edu.cn (J.W.)

**Keywords:** gestational diabetes mellitus, subtypes, gene polymorphism, vitamin D

## Abstract

The present prospective study included 2156 women and investigated the effect of gene variants in the vitamin D (VitD) metabolic and glucose pathways and their interaction with VitD levels during pregnancy on gestational diabetes mellitus (GDM). Plasma 25(OH)D concentrations were measured at the first and second trimesters. GDM subtype 1 was defined as those with isolated elevated fasting plasma glucose; GDM subtype 2 were those with isolated elevated postprandial glucose at 1 h and/or 2 h; and GDM subtype 3 were those with both elevated fasting plasma glucose and postprandial glucose. Six Gc isoforms were categorized based on two *GC* gene variants rs4588 and rs7041, including 1s/1s, 1s/2, 1s/1f, 2/2, 1f/2 and 1f/1f. *VDR*-rs10783219 and *MTNR1B*-rs10830962 were associated with increased risks of GDM and GDM subtype 2; interactions between each other as well as with *CDKAL1*-rs7754840 were observed (*P*_interaction_ < 0.05). Compared with the 1f/1f isoform, the risk of GDM subtype 2 among women with 1f/2, 2/2, 1s/1f, 1s/2 and 1s/1s isoforms and with prepregnancy body mass index ≥24 kg/m^2^ increased by 5.11, 10.01, 10, 14.23, 19.45 times, respectively. Gene variants in VitD pathway interacts with VitD deficiency at the first trimester on the risk of GDM and GDM subtype 2.

## 1. Introduction

Gestational diabetes mellitus (GDM) is a growing public health problem [1,2] and associated with adverse perinatal and neonatal outcomes, including increased risks of gestational hypertension [3], preterm birth [4] and cardiovascular diseases [5]. Although a few risk factors of GDM have been identified, the etiology has not fully been elucidated [6].

Some research has focused on the genetic susceptibility of GDM. Moen et al. [7] found that *MAP3K1*-rs116745876, *PRKCE*-rs11682804 and *NUAK1*-rs11112715 were associated with higher fasting glucose levels at the first trimester and higher 2 h post-oral glucose levels at the second trimester in pregnant women. Other two single nucleotide polymorphisms (SNPs) *CDKAL1*-rs7754840 and *MTNR1B*-rs10830962 identified from a genome-wide association study of GDM were found to be highly correlated with GDM, and another one, *IGF2BP2*-rs1470579, was relatively weakly correlated [8]. On the other hand, genetic variants in the vitamin D (VitD) metabolic pathway were also found to be involved in the pathogenesis of insulin resistance and GDM [9,10,11]. The main circulating metabolite is 25(OH)D, a biomarker of VitD status. VitD metabolism is highly regulated, and variation in the expression or activity of key proteins may modify its level or effects. Key metabolic enzymes include: 25-hydroxylase (*CYP3A4*), which converts VitD to 25(OH)D; 1-hydroxylase (*CYP27B1*), which activates 25(OH)D to 1,25(OH)_2_D; 24-hydroxylase (*CYP24A1*), which inactivates 25(OH)D and 1,25(OH)_2_D; and megalin (*LRP2*), which reabsorbs 25(OH)D through endocytosis in the renal tubules. Other key components include vitamin D-binding protein (*GC*), which transports circulating metabolites, and the VitD receptor (*VDR*), which binds 1,25(OH)_2_D to activate gene transcription and regulates VitD metabolism [12]. Compared to pregnant women with the CC genotype at *VDR*-rs1544410, the risk of GDM in pregnant women with the CT genotype was approximately doubled; compared to AA genotype at *VDR*-rs731236, the risk of GDM in pregnant women with the GA genotype was 1.42 times higher [13]. In addition, two SNPs, rs4588 and rs7041 on the *GC* gene, can form three allelic combinations (Gc1f, Gc1s and Gc2) and six different Gc isoforms, namely, 1s/1s, 1s/2, 1s/1f, 2/2, 1f/2 and 1f/1f [14,15]. According to the free hormone hypothesis, only free 25(OH)D and free 1,25(OH)_2_D can directly exert biological functions [16,17], the proportion of which in blood were mostly influenced by the binding affinity of different Gc isoforms [18]. The polymorphism of VitD metabolic pathway genes, especially on the *GC* genes, may be good candidates to better understand how VitD levels are involved in the pathogenesis of GDM.

Most previous studies have regarded GDM as a homogenous disease, and little attention has been paid to GDM subtypes on the basis of the different time-point glucose levels of the oral glucose tolerance test (OGTT) [8,13]. Studies in non-pregnant women found that both isolated impaired fasting glucose (IFG) and isolated impaired glucose tolerance (IGT) patients were insulin resistance (IR) factors, but the target organs or tissues of IR were different [19,20,21,22]. Individuals with isolated IFG primarily manifest hepatic IR and relatively normal muscle IR. Otherwise, individuals with isolated IGT have normal to subtle hepatic IR and moderate to severe muscle IR. Thus, individuals with both IFG and IGT have both hepatic and muscle IR [19]. The different pathophysiological mechanisms of fasting and post-glycemic abnormalities result from distinct insulin sensitivity characteristics of the liver and muscle, respectively [20,21]. In addition, our previous population-based study found that VitD was associated with the occurrence of GDM with abnormal fasting glucose, especially among overweight/obese pregnant women, but not the occurrence of abnormal post-load glucose [23]. However, previous studies principally treated GDM as a dichotomous outcome when investigating the effects of gene variants on the VitD metabolic and glucose pathways on GDM, ignoring the different pathophysiological mechanisms of fasting and post-load glycemic abnormality [24].

Thus, the aim of this study was to explore the effect of gene variants in the VitD and glucose metabolic-pathway-related genes, and their interactions with 25(OH)D concentrations on the development of GDM and GDM subtypes.

## 2. Materials and Methods

### 2.1. Study Design and Participants

This prospective cohort study was based on the data of Zhoushan Pregnant Women Cohort (ZPWC) from August 2011 to May 2018, which is an ongoing prospective cohort conducted in Zhoushan Maternal and Child Health Care Hospital, Zhejiang. Pregnant women were invited to participate in the cohort at their first prenatal visit. A more detailed description of the inclusion and exclusion criteria has previously been described in detail [23]. Briefly, pregnant women aged between 18 and 45 years without serious physical, mental health disease, threatened abortion or fetal malformation, and who received OGTT were included in the study. Informed consent was obtained from all participants before the investigation.

### 2.2. Collection of Data and Blood Sample

A structured questionnaire was administrated face-to-face by an interviewer to collect information on socio-demographic, lifestyle, and health behavior at the first trimester (T1: 8th–14th gestational week), second trimester (T2: 24th–28th gestational week), third trimester (T3: 32nd–36th gestational week) and 42nd day postpartum. OGTT was conducted during T2 according to a conventional pregnant care program. A 5 mL fasting venous blood sample was drawn at each visit and centrifuged under 4 °C; then, the plasma and white blood cells were divided and stored under −80 °C until use. The results of the OGTT were extracted from the electronic medical records system.

### 2.3. Measurement of 25(OH)D Concentrations

Liquid chromatography–tandem mass spectrometry (API 3200MD (Applied Biosystems/MDS Sciex, Framingham, MA, USA)) was used to measure plasma 25(OH)D_2_ and 25(OH)D_3_ concentrations. The plasma 25(OH)D concentrations were reported in ng/mL, and the lowest sensitivity of the measurement was 2 ng/mL for 25(OH)D_2_ and 5 ng/mL for 25(OH)D_3_. The intra-assay coefficient variance values were 1.47–7.24% and 2.50–7.59% for 25(OH)D_2_ and 25(OH)D_3_, respectively. The inter-assay coefficients variances were 4.48–6.74% and 4.44–6.76% for 25(OH)D_2_ and 25(OH)D_3_, respectively [23]. The 25(OH)D concentrations were the sum of 25(OH)D_2_ and 25(OH)D_3_. The laboratory located in Hangzhou, Zhejiang Province, is CAP-accredited and annually participates in CAP Proficiency Tests and China NCCL Trueness Verification Plan of 25(OH)D Assays, for which satisfactory results in these PT or EQA tests have been obtained in consecutive years.

### 2.4. Covariates Assessment

Plasma 25(OH)D < 20 ng/mL (50 nmol/L) was defined as VitD deficiency according to Endocrine Society clinical practice guidelines [25], and 25(OH)D concentrations ≥20 ng/mL as VitD non-deficiency. Body mass index (BMI) = weight (kg)/height^2^ (m^2^). Prepregnancy BMI was divided into four categories based on the Working Group on Obesity in China [26]: underweight, BMI < 18.5 kg/m^2^; normal, BMI 18.5–23.9 kg/m^2^; overweight, BMI 24.0–27.9 kg/m^2^; obesity, BMI ≥ 28 kg/m^2^. VitD supplementation was categorized as “Yes”, “No” and “Unknown”. According to the sunshine intensity and duration in different months [27], the seasons of blood sampling were divided as follows: spring (March to May), summer (June to August), fall (September to November) and winter (December to February).

### 2.5. GDM and Its Subtypes Classification

GDM screening has become a routine examination among pregnant women in China. OGTT was conducted between the 24th and 28th weeks of gestation. After an overnight fast (at least 8 h), 75 g glucose resolved in 300 mL water was given and drunk within 5 min the next morning. Venous blood samples were taken at 0 h, 1 h and 2 h during OGTT for measuring plasma glucose levels. Plasma glucose levels were immediately measured by the hexokinase method with commercially available kits (Beckman AU5800, Beckman Coulter Inc., Brea, CA, USA). Using criteria proposed by the International Association of the Diabetes and Pregnancy Study Group [28], GDM was diagnosed if any one of the following criteria were met: fasting plasma glucose (FBG) at 0 h ≥5.1 mmol/L, postprandial glucose at 1 h (PG1H) ≥10 mmol/L, or postprandial glucose at 2 h (PG2H) ≥8.5 mmol/L. In addition, according to different types of insulin resistance represented by the blood glucose level at the three time-point glucose levels examined by OGTT [22,23,24], GDM was further categorized into the following three subtypes: GDM subtype 1, with isolated FBG ≥ 5.1 mmol/L; GDM subtype 2, with isolated PG1H ≥ 10 mmol/L and/or PG2H ≥ 8.5 mmol/L; and GDM subtype 3, with both elevated FBG (≥5.1 mmol/L) and post-load plasma glucose (PG1H ≥ 10 mmol/L and/or PG2H ≥ 8.5 mmol/L).

### 2.6. SNP Selection and Genotyping

GDM-related SNP selection: to verify the previous findings by Kwak et al. [8] in Korean pregnant women and Moen et al. [7] among pregnant women in Norway, 3 SNPs (*CDKAL1*-rs7754840, *MTNR1B*-rs10830962 and *IGF2BP2*-rs1470579) related to GDM [8] and 3 SNPs (*MAP3K1*-rs116745876, *PRKCE*-rs11682804 and *NUAK1*-rs11112715) related to blood glucose during pregnancy were selected [7]. According to the minor allele frequency ≥10 of each SNP in the Chinese population from the 1000 Genomes Project database, 4 GDM-related SNPs, *CDKAL1*-rs7754840, *MTNR1B*-rs10830962, *IGF2BP2*-rs1470579 and *PRKCE*-rs11682804, were finally included.

VitD-related SNP selection: the selection conditions of the VitD-related SNP in the study were as follows (satisfy any one) [15]: (1) a positive association between SNP and 25(OH)D concentration reported in the literature, and the minimum allele frequency (Minor allele frequency, MAF) ≥10%; (2) SNPs displayed in the functional region in the NCBI database: exon region, intron splicing point, 5′end and 3′end regulatory regions, and MAF ≥10%; (3) HapMap Chinese database, including gene regions, SNPs within 1500 bp at the 5′end and 3′end, using HaploView to select SNPs, and the conditions are: MAF ≥ 10%; R^2^ ≥ 0.8 [15]. In addition, *VDR* is closely related to insulin secretion [29,30], and *VDR*-rs11568820 is a functional SNP of the *VDR* gene. Previous studies found that rs10783219 and rs11568820 on *VDR* have high LD (r^2^ = 0.98). Therefore, the rs10783219 was selected as the surrogate SNP of rs11568820 [15]. Finally, a total of 13 SNPs related to 25(OH)D concentration in the VitD metabolic pathway were selected (*CYP24A1*: rs2209314, *CYP3A4*: rs2242480, *GC*: rs1155563, rs16846876, rs17467825, rs2282679, rs2298849, rs2298850, rs3755967, rs4588, rs7041, *LRP2*: rs10210408 and *VDR*: rs10783219).

Gc isoforms: based on two SNPs, rs4588 and rs7041, on the *GC* gene, the Gc isoform was categorized into six different isoforms, including 1s/1s, 1s/2, 1s/1f, 2/2, 1f/2 and 1f/1f, of which the proportions of free 25(OH)D were successively reduced. The 1f/1f isoform with the highest proportion of free 25(OH)D was used as the reference group.

The conventional phenol–chloroform extraction method was used to extract DNA from the peripheral blood leukocytes, which was then stored in TE-buffer at −80 °C. For SNP analysis, DNA was then diluted to 10 ng/μL using a Nanodrop^®^ ND-1000 Spectrophotometer (Thermo Fisher Scientific Inc., Wilmington, NC, USA). A Sequenom MassARRAY iPLEX Gold platform (Sequenom, San Diego, CA, USA) was used for SNP genotyping. In total, 17 SNPs were available for further analysis. The call rate of these SNPs was over 98%, which conformed to the Hardy–Weinberg equilibrium.

### 2.7. Statistical Analysis

*t*-tests and Wilcoxon signed-rank tests were used to compare the characteristics between GDM and non-GDM groups for continuous variables. Variance analysis was used to compare the characteristics between different GDM subtypes for continuous variables, and chi-squared tests were used for categorical variables between groups. Multiple linear regression models were used to analyze the association of SNPs in VitD and glucose metabolic-pathway-related genes, and their interactions with 25(OH)D concentrations at T1 and T2 with the blood glucose levels of each OGTT timepoint in a co-dominant genetic model. Multiple logistic regression models were used to analyze the relationship of SNPs, Gc isoforms and their interaction with 25(OH)D concentration at T1 and T2 with GDM as well as its subtypes in a co-dominant genetic model. Furthermore, stratification analysis by prepregnancy BMI was carried out to investigate the association between Gc isoforms and the risk of GDM and its subtypes [23]. To investigate the interaction between *VDR*-rs10783219, *CDKAL1*-rs7754840 and *MTNR1B*-rs10830962 on the risk of GDM and its subtypes, stratification analysis was carried out. In addition, to investigate the joint association of VitD status at T1 or/and T2 with Gc isoforms on the risk of GDM and its subtypes, we classified Gc isoforms into three groups—1f/1f and 1f/2; 2/2 and 1s/1f; and 1s/2 and 1s/1s—and crossover analysis was carried out. The hierarchical analysis was used to investigate the interaction between each SNP and 25(OH)D concentration on the risk of GDM, and the *p*-value of the interaction term was calculated. To investigate whether there was a dose–effect relationship between Gc isoforms and subtypes of GDM, a trend test was applied in the multiple logistic regression model and Gc isoforms were treated as continuous variables for different isoforms (1s/1s, 1s/2, 1s/1f, 2/2, 1f/2 and 1f/1f), of which the proportion of free 25(OH)D was successively reduced. The above multi-factor models were all adjusted for possible confounding, including maternal age, prepregnancy BMI, OGTT season, etc. All test results were considered statistically significant at a value of *p* < 0.05. All analyses were performed using SAS (version 9.2, SAS Institute).

Sample size calculation: in the present study, the risks of GDM subtype 2 of GG genotype in *MTNR1B*-rs10830962 were 1.85 times greater than compared with the CC genotype. The prevalence of GDM in this study was 23.8%; among them, 58.5% were GDM subtype 2. We hypothesized that α = 0.05, power = 80%, OR_gene_ = 1.85, and the genotype frequency for SNP was 18%. Through QUANTO software, it was determined that the minimum case number for the GDM subtype 2 was 118, and the minimum case number for GDM was 202, which is lower than the number of GDM cases in this study (*n* = 513). Therefore, the sample size was large enough for the analysis of different GDM subtypes.

## 3. Results

### 3.1. Subject Characteristics

A total of 2156 pregnant women were included in this study, and the characteristics of the participants are shown in Table 1. Of these, 513 (23.8%) women were diagnosed with GDM. The mean age and prepregnancy BMI of participants were 28.8 years old and 20.7 kg/m^2^, respectively. Compared with non-GDM women, women with GDM had higher prepregnancy BMI, lower 25(OH)D concentrations at T2 and lower educational levels. As shown in Appendix A, compared with participants with GDM subtype 1, those with GDM subtype 2 and 3 were older and had higher VitD levels at T1 and T2.

### 3.2. Associations of SNPs and Its Interaction with VitD on GDM and GDM Subtypes

Compared with the wild-type genotype, the PG1H and/or PG2H levels of mutant genotypes were lower for *LRP2*-rs10210408, and higher for *VDR*-rs10783219, *CDKAL1*-rs7754840 and *MTNR1B*-rs10830962. Interactions between 25(OH)D concentrations at T1 and the CT genotype in *CYP3A4*-rs2242480, GA genotype in *GC*-rs2298849 and CC genotype in *CDKAL1*-rs7754840 on PG1H level, and the CT genotype in *CYP24A1*-rs2209314, TT genotype in *GC*-rs16846876 and GA genotype in *GC*-rs2298849 on PG2H level were observed (Appendix A, all *P*_interaction_ < 0.05). The risks of GDM and GDM subtype 2 of TA genotype in *VDR*-rs10783219 were 1.26 and 1.33 times greater compared with the AA genotype (Table 2). Compared with the CC genotype, GG genotypes in *MTNR1B*-rs10830962 were at higher risk of GDM (Table 2, OR = 2.08, 95% CI: 1.46–2.97), GDM subtype 1 (Table 2, OR = 3.26, 95% CI: 1.62–6.59) and subtype 2 (Table 2, OR = 1.85, 95% CI: 1.22–2.81). Compared with the wild-type genotypes, interactions between 25(OH)D concentrations at T1 and the CT genotype in *CYP3A4*-rs2242480, and the TT genotype in *LRP2*-rs10210408 on the risk of GDM and GDM subtype 2 were found (Table 2). However, interactions between SNPs and 25(OH)D concentrations at T2 on FBG, PG1H and PG2H levels of OGTT as well as GDM and its subtypes were not observed.

As shown in Table 3, significant interactions between *CDKAL1*-rs7754840 and *VDR*-rs10783219 on the risk of GDM and GDM subtype 2 (*P*_interaction_: 0.0121 and 0.0432) as well as interactions between *CDKAL1*-rs7754840 and *MTNR1B*-rs10830962 on the risk of GDM and GDM subtype 1 (*P*_interaction_: 0.0082 and 0.0071) were found.

### 3.3. Associations of Gc Isoforms and VitD with GDM and GDM Subtypes

Compared to women with Gc isoforms of 1f/1f, those with Gc isoforms of 2/2 and 1s/2 had higher levels of PG1H and PG2H among women with prepregnancy BMI ≥ 24 kg/m^2^ (Appendix A). In addition, after adjusting for potential confounders, dose–effect relationships of Gc isoforms with GDM and GDM subtype 2 (*P*_trend_: 0.0046 and 0.0011, Appendix A) were observed among women with prepregnancy BMI ≥ 24 kg/m^2^. Compared to women with Gc isoforms of 1f/1f and 1f/2 and VitD non-deficiency at T1 and T2, those with Gc isoforms of 1s/2 and 1s/1s had increased risk of GDM and GDM subtype 2 (OR = 2.21, 95% CI: 1.14–4.30; OR = 2.79, 95% CI: 1.20–6.49, Table 4). However, combined effect of 25(OH)D concentrations at T1 or T2 with Gc isoforms on the risk of GDM and GDM subtypes were not observed (Table 4).

## 4. Discussion

The current study demonstrated significant associations of variant genotype of SNPs at *VDR*-rs10783219 and *MTNR1B*-rs10830962 with the risk of GDM and GDM subtype 2. Furthermore, *CDKAL1*-rs7754840 interacts with *VDR*-rs10783219 and *MTNR1B*-rs10830962 on GDM subtypes. In addition, among women with prepregnancy BMI ≥ 24 kg/m^2^, a dose–effect relationship between Gc isoforms and GDM subtype 2 was observed.

The *LRP2* gene plays an important role in the preservation of vitamin D metabolites and delivery of the precursor to the kidney for the generation of 1α,25(OH)_2_D_3_ [15,31], polymorphisms of which were associated with increased risks of severe VitD deficiency and related bone disease [32]. Our study initially found that variation at *LRP2*-rs10210408 was related to higher postprandial glucose levels among pregnant women. In addition, interactions between *LRP2*-rs10210408 and VitD level at T1 on the risk of GDM and GDM subtype 2 were found, which indicated that variations of the A allele to T at *LRP2*-rs10210408 might influence glucose metabolism through VitD during pregnancy.

*VDR*-rs11568820 is a functional SNP and its variant may improve the islet activity of the calcium-sensing receptor, which further inhibits insulin secretion [33]. Only one study has reported that the variant at *VDR*-rs11568820 impairs the secretion of pancreatic islets and increases the risk of type 2 diabetes in the adult cohort and PG2H in children [30]. In the present study, we identified that the homozygous variant at *VDR*-rs10783219 in pregnant women was associated with higher PG1H (β = 0.24, *p* = 0.0212), and higher risks of GDM (TA/TT vs. AA: OR = 1.28) and GDM subtype 2 (TA/TT vs. AA: OR = 1.31). According to the high-linkage relationship between *VDR*-rs10783219 and *VDR*-rs11568820 in this population [15], we could speculate that it might be the highly interlinked *VDR*-rs11568820 that exhibits the biological functions. *VDR*-rs11568820 not only plays an important role in the development of type 2 diabetes, but also of GDM. Significant associations between *CDKAL1*-rs7754840 and PG2H, as well as GDM, were also observed in our study, which was consistent with the genome-wide association study reported by Kwak et al. [8]. Variants at *CDKAL1*-rs7754840 may affect the conversion process from proinsulin to insulin [34]. This study further confirmed that variants at *CDKAL1*-rs7754840 increased the risk of GDM in Chinese populations. Furthermore, we also found that for each additional G risk allele at *MTNR1B*-rs10830962, the risk of GDM increased by 52% and 108%, and GDM subtype 2 by 43% and 85%, respectively, which was consistent with previous studies [8,35]. The *MTNR1B* gene encodes melatonin receptor 2 (MTNR2), which could significantly inhibit the expression of 3′5′-cyclic adenosine monophosphate in cells, and subsequently reduces insulin secretion [36,37]. Therefore, variants of the C allele to G at *MTNR1B*-rs10830962 are likely to inhibit the release of insulin in islet cells and increase the risk of GDM.

Meanwhile, we also identified a significant interaction between *VDR*-rs10783219 and *CDKAL1*-rs7754840 as well as *MTNR1B*-rs10830962 on GDM. Variants at *VDR*-rs10783219 increased the risk of GDM and GDM subtype 2 among women with a variant at *CDKAL1*-rs7754840, suggesting that the protective effect of VitD on GDM was more obvious in patients with abnormal islet cell functions. In addition, the T allele at *VDR*-rs10783219 and the C allele at *CDKAL1*-rs7754840 separately increased the risk of GDM subtype 1 among women with the GG genotype at *MTNR1B*-rs10830962 (OR = 2.99, 95%CI: 1.34–6.68; OR = 3.06, 95%CI: 1.41–6.66) (*P*_interaction_ = 0.2611; *P*_interaction_ = 0.0071). Given that the *MTNR1B* gene could reduce the secretion of insulin, the conversion obstacles of proinsulin to insulin mediated by the *CDKAL1* gene might be strengthened with reduced insulin secretion. The above interaction between SNPs found in this study provides a new perspective for the study of the pathogenesis of GDM, but the specific biological mechanism still needs to be verified by further studies.

Traditionally, 25(OH)D was thought to be taken up by cells of the kidney binding to vitamin D-binding protein through megalin/cubilin-mediated endocytosis. However, studies [38,39] have found that although the levels of both 25(OH)D and 1,25(OH)_2_D in blood and urine were low in megalin knockout and vitamin D-binding protein knockout mice, vitamin D-binding protein knockout mice did not show symptoms of VitD deficiency, unlike megalin knockout mice. In addition, vitamin D-binding protein knockout mice would rapidly manifest symptoms of VitD deficiency when fed with a VitD-deficient diet. In 2019, the first case of the human homozygous deletion of a *GC* gene reported by Henderson et al. [17] confirmed that this mechanism found in animals also applies to humans. The above research indicates that free 25(OH)D or 1,25(OH)_2_D is the main form to exert the biological VitD effects. Furthermore, the proportion of free 25(OH)D of individuals with different Gc isoforms is different: individuals with the 1f/1f isoform have the highest free 25(OH)D concentrations, and individuals with 1s/1s have the lowest, followed by 1f/2, 2/2, 1s/1f and 1s/2 [16]. This study initially reported that the associations of Gc isoforms with GDM and GDM subtypes during pregnancy were different in pregnant women with different prepregnancy BMI. Significant associations were only observed among women who were overweight or obese before pregnancy. The distribution of Gc isoforms was significantly different between blacks and whites along with the distribution of fat with the same BMI [40]. More than 90% of blacks were of Gc1f type, whereas the majority of whites are of Gc1s type; Asians were in between [41]. The accumulation of abdominal fat is a risk factor for insulin resistance and metabolic syndrome [42]. Given the strong association between BMI and insulin resistance [43], we speculated that overweight and obese pregnant women might have underlying insulin resistance before pregnancy, and the difference in insulin resistance among pregnant women with different Gc isoforms may be caused by the difference in body fat distribution. In this study, it was found that compared with the 1f/1f isoform, pregnant women with 1s/2 and 1s/1s isoforms had higher risk of GDM subtype 2, indicating higher visceral and liver fat content, and thus, higher muscle insulin resistance. However, the specific pathophysiological mechanism needs to be confirmed by further studies.

Our previous study [23] found that serum 25(OH)D only affected FBG and GDM subtypes with abnormal fasting glucose. However, this study found that free 25(OH)D (represented by Gc isoforms) mainly influences postprandial glucose levels and GDM subtype 2. The difference between serum 25(OH)D and free 25(OH)D on glucose and GDM risk indicates that the proportion of free 25(OH)D is mainly related to muscle insulin resistance or insulin secretion, and serum 25(OH)D in circulation is not mainly mediated by free 25(OH)D, which may be related to fasting gluconeogenesis levels in the liver, and plays its role in lowering glucose levels through megalin/cubilin-mediated endocytosis through the kidney or parathyroid cells [44]. However, combined effects of 25(OH)D concentrations at T1 or T2 with Gc isoforms on the risk of GDM and GDM subtypes were not observed.

Strengths of the current study included the prospective cohort design and the relatively large sample size, which may guarantee the authenticity of the research results and higher statistical test efficiency. Furthermore, we initially divided GDM into different subtypes based on the different mechanisms of insulin resistance. The risks of GDM and GDM subtypes in pregnant women with different Gc isoforms have been investigated for the first time, and the effect of prepregnancy BMI and longitudinal changes in VitD during pregnancy on the association between Gc isoforms and GDM as well as its subtypes was considered. However, there were several potential limitations in this study. Insulin levels, which could more accurately distinguish different types of insulin resistance in GDM, were not detected simultaneously during the OGTT examination in this study. In addition, the average prepregnancy BMI of the population in this study was low, and about 12% of the pregnant women were overweight (10.3%) or obese (2.1%). Furthermore, in this study, we investigated whether there was a VitD supplementation of participants during pregnancy, but did not consider the supplementation dose because the clinically recommended supplementation dose of VitD for pregnant women is between 400 and 600 IU. However, the type of VitD supplementation was unknown, which restricted the study to further explore how the SNP affected the response to VitD supplementation on serum 25(OH)D concentrations and its impact on GDM. Therefore, the results of this study may be limited when extrapolating to obese or severely obese pregnant women.

## 5. Conclusions

In conclusion, our results showed that variants of SNPs at *VDR*-rs10783219 and *MTNR1B*-rs10830962 significantly increased the risk of GDM and GDM subtypes with normal fasting glucose and elevated post-load glucose, and interactions were investigated between each other as well as with *CDKAL1*-rs7754840. With lower Gc isoforms, the proportions of free 25(OH)D were related to an increased risk of GDM with abnormal postprandial blood glucose in prepregnancy overweight and obese women. The present study explored whether gene variants in the VitD metabolic and glucose pathway would affect the risk of GDM from a genetic point of view. In addition, the 25(OH)D concentration is very unstable and can easily be affected by exposure factors such as supplementation and sunlight exposure. Identifying the effect of gene variants in the VitD and glucose metabolic-pathway-related genes on the development of GDM and GDM subtypes could more objectively evaluate the relationship between VitD and GDM and provide standards for subsequent clinical applications.

## Figures and Tables

**Table 1 nutrients-13-04220-t001:** Baseline characteristics of pregnant women.

Variables	Total	non-GDM	GDM	*p*
	*n* = 2156	*n* = 1643	*n* = 513	
Age, years	28.8 (3.7)	28.5 (3.5)	29.6 (4.0)	<0.0001
Prepregnancy BMI (kg/m^2^)	20.7 (2.8)	20.6 (2.7)	21.3 (3.0)	<0.0001
25(OH)D at T1 (ng/mL) *	18.9 (8.7)	18.7 (8.7)	19.5 (8.6)	0.0884
25(OH)D_3_	18.1 (8.6)	17.9 (8.6)	18.7 (8.6)	0.0530
25(OH)D_2_ ^¶^	0.6 (0.5)	0.6 (0.5)	0.5 (0.4)	0.9761 ^⁋^
25(OH)D at T2 (ng/mL) ^†^	25.6 (11.5)	26.0 (11.7)	24.1 (10.7)	0.0149
25(OH)D_3_	24.6 (11.5)	24.9 (11.7)	23.3 (10.8)	0.0310
25(OH)D_2_ ^¶^	0.7 (0.7)	0.6 (0.7)	0.7 (0.6)	0.6255 ^⁋^
VitD deficiency at T1 *	1281 (62.3%)	983 (62.8%)	298 (60.7%)	0.3979
VitD deficiency at T2 ^†^	499 (36.4%)	374 (34.8%)	125 (42.2%)	0.0180
GDM rate	513 (23.8%)	—	—	—
OGTT season				0.0920
Summer/fall	1045 (48.5%)	813 (49.5%)	232 (45.2%)	
Winter/spring	1111 (51.5%)	830 (50.5%)	281 (54.8%)	
Educational level				0.0179
≤High school	589 (27.3%)	428 (26.0%)	161 (31.4%)	
>High school	1567 (72.7%)	1215 (74.0%)	352 (68.6%)	
Income per capita, RMB				0.3659
<30,000	191 (8.9%)	143 (8.7%)	48 (9.4%)	
≥30,000	1647 (76.4%)	1269 (77.2%)	378 (73.7%)	
Not sure	180 (8.3%)	132 (8.0%)	48 (9.4%)	
Unknown	138 (6.4%)	99 (6.0%)	39 (7.6%)	
Planned pregnancy				0.0411
No	709 (32.9%)	563 (34.3%)	146 (28.5%)	
Yes	1313 (60.9%)	983 (59.8%)	330 (64.3%)	
Unknown	134 (6.2%)	97 (5.9%)	37 (7.2%)	
Marital status				0.5033
Not married	47 (2.2%)	35 (2.1%)	12 (2.3%)	
Married	1976 (91.7%)	1512 (92.0%)	464 (90.4%)	
Unknown	133 (6.2%)	96 (5.8%)	37 (7.2%)	
VitD supplement				0.4623
0/week	765 (35.5%)	593 (36.1%)	172 (33.5%)	
>0/week	1233 (57.2%)	934 (56.8%)	299 (58.3%)	
Unknown	158 (7.3%)	116 (7.1%)	42 (8.2%)	
Primiparity				0.1854
No	491 (22.8%)	359 (21.9%)	132 (25.7%)	
Yes	1498 (69.5%)	1156 (70.4%)	342 (66.7%)	
Unknown	167 (7.7%)	128 (7.8%)	39 (7.6%)	
Physical exercise				0.0775
0/week	1717 (79.6%)	1326 (80.7%)	391 (76.2%)	
>0/week	292 (13.5%)	213 (13.0%)	79 (15.4%)	
Unknown	147 (6.8%)	104 (6.3%)	43 (8.4%)	

Abbreviations: GDM, gestational diabetes mellitus; VitD, vitamin D; T1, first trimester; T2, second trimester; OGTT, oral glucose tolerance test. * *n* = 2056, ^†^
*n* = 1372, ^¶^ Presented as the median (interquartile range), ^⁋^ compared by Wilcoxon signed-rank test.

**Table 2 nutrients-13-04220-t002:** Relationship of SNPs in VitD and glucose metabolic pathway and its interaction with 25(OH)D concentrations at T1 and T2 with GDM and GDM subtypes *.

SNPs	Genotypes	*n*	GDM ^†^		GDM Subtype 1 ^‡^		GDM Subtype 2 ^‡^		GDM Subtype 3 ^‡^	
			Case (%)	OR (95% CI)	Case (%)	OR (95% CI)	Case (%)	OR (95% CI)	Case (%)	OR (95% CI)
VitD-related SNPs										
*CYP24A1*										
rs2209314	TT	770	193 (25.1)	Ref	62 (8.1)	Ref	106 (13.8)	Ref	25 (3.2)	Ref
	CT	1039	244 (23.5)	0.93 (0.75–1.17)	62 (6.0)	0.71 (0.49–1.04)	151 (14.5)	1.07 (0.81–1.42)	31 (3.0)	0.92 (0.53–1.61)
	CC	335	75 (22.4)	0.86 (0.63–1.17)	20 (6.0)	0.72 (0.42–1.23)	42 (12.5)	0.89 (0.60–1.33)	13 (3.9)	1.10 (0.54–2.27)
*CYP3A4*										
rs2242480	CC	1229	292 (23.8)	Ref	85 (6.9)	Ref	170 (13.8)	Ref	37 (3.0)	Ref
	CT	790	191 (24.2)	1.04 (0.84–1.29) ^||^	54 (6.8)	1.13 (0.78–1.63)	109 (13.8)	0.97 (0.74–1.26) ^||^	28 (3.5)	1.32 (0.78–2.23)
	TT	125	28 (22.4)	0.96 (0.61–1.51)	5 (4.0)	0.59 (0.23–1.53)	19 (15.2)	1.08 (0.63–1.82)	4 (3.2)	1.30 (0.43–3.92)
*GC*										
rs1155563	TT	761	169 (22.2)	Ref	47 (6.2)	Ref	95 (12.5)	Ref	27 (3.5)	Ref
	TC	1019	248 (24.3)	1.15 (0.91–1.44)	72 (7.1)	1.16 (0.78–1.72)	146 (14.3)	1.20 (0.90–1.60)	30 (2.9)	0.90 (0.52–1.57)
	CC	362	94 (26.0)	1.19 (0.88–1.60)	25 (6.9)	1.13 (0.67–1.89)	57 (15.7)	1.29 (0.89–1.86)	12 (3.3)	0.89 (0.43–1.84) ^||^
rs16846876	AA	1017	229 (22.5)	Ref	63 (6.2)	Ref	126 (12.4)	Ref	40 (3.9)	Ref
	AT	899	220 (24.5)	1.09 (0.87–1.35)	62 (6.9)	1.10 (0.76–1.60)	137 (15.2)	1.25 (0.95–1.63)	21 (2.3)	0.54 (0.31–0.94)
	TT	231	61 (26.4)	1.17 (0.84–1.64)	19 (8.2)	1.31 (0.75–2.29)	34 (14.7)	1.25 (0.82–1.90)	8 (3.5)	0.78 (0.34–1.75)
rs17467825	AA	1008	228 (22.6)	Ref	61 (6.1)	Ref	132 (13.1)	Ref	35 (3.5)	Ref
	GA	909	224 (24.6)	1.10 (0.89–1.36)	66 (7.3)	1.15 (0.79–1.66)	132 (14.5)	1.15 (0.88–1.50)	26 (2.9)	0.79 (0.46–1.35)
	GG	234	59 (25.2)	1.08 (0.77–1.51)	16 (6.8)	1.05 (0.58–1.89)	35 (15.0)	1.18 (0.78–1.80)	8 (3.4)	0.81 (0.36–1.85)
rs2282679	TT	1009	227 (22.5)	Ref	61 (6.0)	Ref	130 (12.9)	Ref	36 (3.6)	Ref
	GT	899	224 (24.9)	1.13 (0.91–1.40)	67 (7.5)	1.18 (0.81–1.71)	132 (14.7)	1.20 (0.92–1.57)	25 (2.8)	0.75 (0.44–1.29)
	GG	241	60 (24.9)	1.07 (0.76–1.50)	16 (6.6)	1.00 (0.55–1.81)	36 (14.9)	1.20 (0.80–1.82)	8 (3.3)	0.77 (0.34–1.74)
rs2298849	AA	894	216 (24.2)	Ref	66 (7.4)	Ref	120 (13.4)	Ref	30 (3.4)	Ref
	GA	960	231 (24.1)	1.03 (0.83–1.28)	58 (6.0)	0.82 (0.56–1.20)	145 (15.1)	1.16 (0.88–1.52)	28 (2.9)	0.93 (0.54–1.61)
	GG	299	65 (21.7)	0.87 (0.63–1.19)	20 (6.7)	0.89 (0.52–1.51)	34 (11.4)	0.80 (0.53–1.21)	11 (3.7)	1.08 (0.52–2.26)
rs2298850	GG	982	221 (22.5)	Ref	60 (6.1)	Ref	127 (12.9)	Ref	34 (3.5)	Ref
	CG	911	227 (24.9)	1.13 (0.91–1.40)	67 (7.4)	1.17 (0.81–1.70)	134 (14.7)	1.19 (0.91–1.55)	26 (2.9)	0.80 (0.47–1.38)
	CC	240	59 (24.6)	1.06 (0.75–1.48)	16 (6.7)	0.99 (0.55–1.80)	35 (14.6)	1.17 (0.77–1.78)	8 (3.3)	0.81 (0.36–1.83)
rs3755967	CC	1005	226 (22.5)	Ref	61 (6.1)	Ref	130 (12.9)	Ref	35 (3.5)	Ref
	CT	907	226 (24.9)	1.14 (0.91–1.41)	67 (7.4)	1.17 (0.81–1.70)	133 (14.7)	1.20 (0.91–1.57)	26 (2.9)	0.80 (0.47–1.37)
	TT	241	60 (24.9)	1.07 (0.76–1.50)	16 (6.6)	1.00 (0.55–1.80)	36 (14.9)	1.20 (0.79–1.81)	8 (3.3)	0.79 (0.35–1.79)
rs4588	GG	994	226 (22.7)	Ref	61 (6.1)	Ref	129 (13.0)	Ref	36 (3.6)	Ref
	GT	909	226 (24.9)	1.11 (0.89–1.38)	67 (7.4)	1.15 (0.80–1.67)	134 (14.7)	1.18 (0.90–1.55)	25 (2.8)	0.73 (0.42–1.25)
	TT	241	59 (24.5)	1.02 (0.73–1.44)	16 (6.6)	0.97 (0.54–1.75)	35 (14.5)	1.14 (0.75–1.73)	8 (3.3)	0.75 (0.33–1.70)
rs7041	AA	1162	271 (23.3)	Ref	79 (6.8)	Ref	153 (13.2)	Ref	39 (3.4)	Ref
	CA	826	201 (24.3)	1.08 (0.87–1.34)	57 (6.9)	1.07 (0.74–1.54)	119 (14.4)	1.13 (0.86–1.47)	25 (3.0)	0.93 (0.55–1.59)
	CC	162	41 (25.3)	1.22 (0.82–1.79)	8 (4.9)	0.89 (0.42–1.93)	28 (17.3)	1.38 (0.87–2.18)	5 (3.1)	1.25 (0.46–3.35)
*LRP2*										
rs10210408	CC	703	181 (25.7)	Ref	45 (6.4)	Ref	110 (15.6)	Ref	26 (3.7)	Ref
	TC	1065	229 (21.5)	0.78 (0.62–0.99)	67 (6.3)	0.92 (0.61–1.37)	132 (12.4)	0.73 (0.55–0.97)	30 (2.8)	0.77 (0.44–1.34)
	TT	385	102 (26.5)	1.07 (0.80–1.43) ^||^	32 (8.3)	1.32 (0.81–2.16)	57 (14.8)	0.97 (0.68–1.39) ^||^	13 (3.4)	1.09 (0.54–2.21)
*VDR*										
rs10783219	AA	809	173 (21.4)	Ref	51 (6.3)	Ref	100 (12.4)	Ref	22 (2.7)	Ref
	TA	1010	254 (25.1)	1.26 (1.00–1.58) ^§^	67 (6.6)	1.08 (0.73–1.60)	154 (15.2)	1.33 (1.01–1.76) ^§^	33 (3.3)	1.32 (0.74–2.33)
	TT	332	86 (25.9)	1.32 (0.98–1.80)	26 (7.8)	1.35 (0.81–2.25)	46 (13.9)	1.24 (0.84–1.82)	14 (4.2)	1.66 (0.81–3.41)
rs10783219	AA	809	173(21.4)	Ref	51 (6.3)	Ref	100 (12.4)	Ref	22 (2.7)	Ref
	TA/TT	1342	340(25.3)	1.28 (1.03–1.58) ^§^	93 (6.9)	1.15 (0.80–1.65)	200 (14.9)	1.31 (1.01–1.71) ^§^	47 (3.5)	1.40 (0.82–2.40)
GDM-related SNPs										
*CDKAL1*										
rs7754840	GG	635	128 (20.2)	Ref	24 (3.8)	Ref	85 (13.4)	Ref	19 (3.0)	Ref
	GC	820	164 (20.0)	0.99 (0.76–1.29)	31 (3.8)	1.11 (0.63–1.95)	111 (13.5)	1.01 (0.74–1.38)	22 (2.7)	0.89 (0.46–1.70)
	CC	264	63 (23.9)	1.35 (0.95–1.92)	12 (4.5)	1.40 (0.67–2.91)	39 (14.8)	1.25 (0.82–1.91)	12 (4.5)	1.82 (0.84–3.95)
rs7754840	GG/GC	1455	292 (20.1)	Ref	55(3.8)	Ref	196(13.5)	Ref	41(2.8)	Ref
	CC	264	63 (23.9)	1.43 (1.03–1.97) ^§^	12 (4.5)	1.32 (0.68–2.56)	39 (14.8)	1.24 (0.85–1.83)	12 (4.5)	1.94 (0.97–3.88)
*IGF2BP2*										
rs1470579	AA	966	203 (21.0)	Ref	39 (4.0)	Ref	136 (14.1)	Ref	28 (2.9)	Ref
	CA	664	133 (20.0)	0.95 (0.73–1.22)	25 (3.8)	0.94 (0.55–1.59)	85 (12.8)	0.89 (0.66–1.20)	23 (3.5)	1.25 (0.70–2.24)
	CC	89	18 (20.2)	0.96 (0.55–1.66)	3 (3.4)	0.87 (0.26–2.96)	13 (14.6)	1.02 (0.54–1.92)	2 (2.2)	0.83 (0.19–3.76)
*MTNR1B*										
rs10830962	CC	572	91 (15.9)	Ref	17 (3.0)	Ref	62 (10.8)	Ref	12 (2.1)	Ref
	GC	850	186 (21.9)	1.52 (1.14–2.03) ^§^	30 (3.5)	1.45 (0.77–2.72)	122 (14.4)	1.43 (1.02–2.00) ^§^	34 (4.0)	2.38 (1.18–4.81) ^§^
	GG	297	78 (26.3)	2.08 (1.46–2.97) ^§^	20 (6.7)	3.26 (1.62–6.59) ^§^	51 (17.2)	1.85 (1.22–2.81) ^§^	7 (2.4)	1.83 (0.68–4.88)
*PRKCE*										
rs11682804	GG	839	158 (18.8)	Ref	30 (3.6)	Ref	106 (12.6)	Ref	22 (2.6)	Ref
	AG	745	166 (22.3)	1.21 (0.94–1.56)	29 (3.9)	1.17 (0.69–2.01)	112 (15.0)	1.23 (0.91–1.66)	25 (3.4)	1.26 (0.69–2.31)
	AA	138	31 (22.5)	1.22 (0.78–1.90)	8 (5.8)	1.60 (0.69–3.71)	17 (12.3)	0.95 (0.54–1.69)	6 (4.3)	1.91 (0.73–4.98)

Abbreviations: GDM, gestational diabetes mellitus; VitD, vitamin D; subtype 1, elevated fasting glucose and normal post-load glucose; subtype 2, normal fasting glucose and elevated post-load glucose; subtype 3, elevated fasting and post-load glucose. * Adjusted for maternal age, prepregnancy BMI, parity, educational level, income, physical exercise and OGTT season. ^†^ Binomial logistic regression model; ^‡^ multinomial logistic regression model. ^§^
*p* < 0.05; ^||^
*p*-value of the interaction term SNPs * 25(OH)D concentration at the first trimester < 0.05.

**Table 3 nutrients-13-04220-t003:** Interactions between *CDKAL1*, *MTNR1B* and *VDR* on risk of GDM and GDM subtypes *.

SNPs	Risk Allele of GDM	*n*	GDM ^†^		GDM Subtype 1 ^‡^		GDM Subtype 2 ^‡^		GDM Subtype 3 ^‡^	
			Case (%)	OR (95% CI)	Case (%)	OR (95% CI)	Case (%)	OR (95% CI)	Case (%)	OR (95% CI)
*CDKAL1*-rs7754840	*VDR*-rs10783219									
GG	T	633	128 (20.2)	0.81 (0.60–1.09)	24 (3.8)	0.80 (0.43–1.50)	85 (13.4)	0.76 (0.53–1.08)	19 (3.0)	1.05 (0.52–2.10)
GC	T	819	164 (20.0)	1.35 (1.05–1.75) ^§^	31 (3.8)	1.55 (0.91–2.63)	111 (13.6)	1.34 (1.00–1.81) ^§^	22 (2.7)	1.14 (0.61–2.13)
CC	T	264	63 (23.8)	1.36 (0.89–2.08)	12 (4.6)	1.36 (0.52–3.59)	39 (14.8)	1.16 (0.69–1.95)	12 (4.6)	2.82 (0.99–8.04)
GC/CC	T	1083	227 (21.0)	1.37 (1.10–1.70) ^§^	43 (4.0)	1.51 (0.95–2.38)	150 (13.9)	1.31 (1.02–1.70) ^§^	34 (3.1)	1.49 (0.90–2.44)
			*P*_interaction_ = 0.0121		*P*_interaction_ = 0.2036		*P*_interaction_ = 0.0432		*P*_interaction_ = 0.1768	
*MTNR1B*-rs10830962	*VDR*-rs10783219									
CC	T	572	91 (15.9)	1.26 (0.91–1.76)	17 (3.0)	1.35 (0.64–2.87)	62 (10.8)	1.27 (0.85–1.88)	12 (2.1)	1.31 (0.54–3.18)
GC	T	848	186 (21.9)	0.90 (0.71–1.16)	30 (3.5)	0.72 (0.40–1.27)	122 (14.4)	0.89 (0.67–1.19)	34 (4.0)	1.15 (0.69–1.94)
GG	T	296	78 (26.4)	1.88 (1.20–2.94) ^§^	20 (6.8)	2.99 (1.34–6.68) ^§^	51 (17.2)	1.59 (0.94–2.69)	7 (2.4)	1.70 (0.50–5.76)
			*P*_interaction_ = 0.5882		*P*_interaction_ = 0.2611		*P*_interaction_ = 0.9631		*P*_interaction_ = 0.8731	
*MTNR1B*-rs10830962	*CDKAL1*-rs7754840									
CC	C	572	91 (15.9)	0.89 (0.63–1.24)	17 (3.0)	0.77 (0.37–1.61)	62 (10.8)	0.97 (0.65–1.45)	12 (2.1)	0.74 (0.31–1.74)
GC	C	848	186 (21.9)	1.08 (0.84–1.38)	30 (3.5)	0.86 (0.48–1.53)	122 (14.4)	1.08 (0.81–1.44)	34 (4.0)	1.32 (0.78–2.24)
GG	C	297	78 (26.3)	1.89 (1.23–2.91) ^§^	20 (6.7)	3.06 (1.41–6.66) ^§^	51 (17.2)	1.48 (0.90–2.46)	7 (2.4)	3.66 (0.94–14.26)
			*P*_interaction_ = 0.0082		*P*_interaction_ = 0.0071		*P*_interaction_ = 0.1849		*P*_interaction_ = 0.0653	

Abbreviations: GDM, gestational diabetes mellitus; subtype 1, elevated fasting glucose and normal post-load glucose; subtype 2, normal fasting glucose and elevated post-load glucose; subtype 3, elevated fasting and post-load glucose. * Adjusted for maternal age, prepregnancy BMI, parity, educational level, income, physical exercise and OGTT season. ^†^ Binomial logistic regression model; ^‡^ multinomial logistic regression model. ^§^
*p* < 0.05.

**Table 4 nutrients-13-04220-t004:** The relationship of VitD status at T1 and T2, Gc isoforms with GDM and GDM subtypes *.

VitD Deficiency		Gc Isoforms	*n*	GDM ^a^		GDM Subtype 1 ^b^		GDM Subtype 2 ^b^		GDM Subtype 3 ^b^	
T1	T2			Case (%)	OR (95% CI)	Case (%)	OR (95% CI)	Case (%)	OR (95% CI)	Case (%)	OR (95% CI)
No	No	1f/1f and 1f/2	148	24 (16.2)	Ref	7 (4.7)	Ref	12 (8.1)	Ref	5 (3.4)	Ref
		2/2 and 1s/1f	116	17 (14.7)	0.98 (0.49–1.95)	4 (3.5)	0.73 (0.20–2.60)	9 (7.8)	1.12 (0.44–2.82)	4 (3.5)	1.13 (0.28–4.56)
		1s/2 and 1s/1s	85	23 (27.1)	2.21 (1.14–4.30) ^⁋^	3 (3.5)	1.02 (0.25–4.14)	15 (17.7)	2.79 (1.20–6.49) ^⁋^	5 (5.9)	2.55 (0.66–9.92)
No	Yes	1f/1f and 1f/2	31	11 (35.5)	2.91 (1.19–7.14) ^⁋^	6 (19.4)	4.31 (1.23–15.05) ^⁋^	2 (6.5)	1.31 (0.26–6.58)	3 (9.7)	3.83 (0.74–19.89)
		2/2 and 1s/1f	27	7 (25.9)	2.16 (0.80–5.84)	3 (11.1)	2.19 (0.50–9.71)	3 (11.1)	2.37 (0.59–9.56)	1 (3.7)	1.69 (0.17–17.04)
		1s/2 and 1s/1s	26	5 (19.2)	1.36 (0.46–4.05)	5 (19.2)	3.39 (0.95–12.09)	0 (0.0)	——	0 (0.0)	——
Yes	No	1f/1f and 1f/2	195	43 (22.05)	1.94 (1.09–3.45) ^⁋^	12 (6.2)	1.89 (0.71–5.08)	27 (13.9)	2.27 (1.07–4.82) ^⁋^	4 (2.1)	1.24 (0.30–5.14)
		2/2 and 1s/1f	163	33 (20.3)	1.67 (0.92–3.06)	10 (6.1)	1.83 (0.66–5.08)	21 (12.9)	1.95 (0.89–4.25)	2 (1.2)	0.60 (0.11–3.37)
		1s/2 and 1s/1s	100	21 (21.0)	1.87 (0.95–3.67)	4 (4.0)	1.28 (0.36–4.65)	17 (17.0)	2.56 (1.11–5.87) ^⁋^	0 (0.0)	——
Yes	Yes	1f/1f and 1f/2	165	35 (21.2)	1.71 (0.94–3.11)	22 (13.3)	3.04 (1.21–7.61) ^⁋^	8 (4.9)	0.80 (0.30–2.11)	5 (3.0)	1.57 (0.41–6.07)
		2/2 and 1s/1f	113	31 (27.4)	2.27 (1.22–4.22) ^⁋^	16 (14.2)	3.59 (1.38–9.33) ^⁋^	12 (10.6)	1.81 (0.75–4.36)	3 (2.7)	1.40 (0.31–6.38)
		1s/2 and 1s/1s	95	22 (23.2)	1.84 (0.94–3.60)	9 (9.5)	2.29 (0.79–6.58)	8 (8.4)	1.37 (0.51–3.64)	5 (5.3)	2.59 (0.67–10.02)

Abbreviations: GDM, gestational diabetes mellitus; VitD, vitamin D; T1, first trimester; T2, second trimester; subtype 1, elevated fasting glucose and normal post-load glucose; subtype 2, normal fasting glucose and elevated post-load glucose; subtype 3, elevated fasting and post-load glucose. * Adjusted for maternal age, prepregnancy BMI, parity, educational level, income, physical exercise and OGTT season. ^a^ Binomial logistic regression model; ^b^ multinomial logistic regression model. ^⁋^
*p* < 0.05.

## Data Availability

The data presented in this study are available on request from the corresponding author. The data are not publicly available because they contain information that could compromise the privacy of research participants.

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
