# Peer review of "The Association of Gene Variants in the Vitamin D Metabolic Pathway and Its Interaction with Vitamin D on Gestational Diabetes Mellitus: A Prospective Cohort Study"

_nutrients, 2021, doi:10.3390/nu13124220_

Round 1
Reviewer 1 Report
This is a manuscript describing a pregnancy cohort in regard to GDM and various SNPs pertaining to glucose homeostasis or vitamin D metabolism. The 3 subtypes of GDM is a potentially more novel approach rather than combining all types together; but the data is not really shown according to subgroups, how these subgroups compare for general characteristics and vitamin D status is not known. The manuscript needs some considerations as listed below. Some attention to readability is also needed, e.g. describe the SNPs earlier regarding what these code for so that all readers are able to appreciate the work.
Specific comments:
Title: vitamin D does not need to be capitalized.
Abstract
The 3 subtypes of GDM should be defined.
When referring to concentrations of 25-hydroxyvitamin D use concentrations, not levels.
VitD is not really a suitable abbreviation.
Introduction
I am not sure all readers will know what proteins the genes encode for, this could be added to the introduction in brief. The strength of the various associations is not clear, nor the direction.
The details on IFG and IGT and both apply to non-pregnancy people, how valid this is in pregnancy needs to be provided or some context as to which is most common.
Methods
The cohort study is described in brief.
Line 84, well-trained is not really necessary to state; it is subjective.
Lines 93-100; the sensitivity is reported as lower for the D2 isoform than D3 which is very unusual. Is this correct? In addition the accuracy against NIST SRM is needed; participation in VDSCP or DEQAS should be provided.
Line 102, is citing the IOM, use their definition of deficiency which is <12 ng/mL not 20. IOM defines sufficient as >20 ng/mL.
Lines 106-109, is there a citation to add to back up the information on seasons?
It is not clear in the statistical section how many women needed to be recruited to have enough with GDM of the various subtypes. A sample size estimate is needed.
Results
Table 1, the difference between non-GDM and GDM for VitD at T2 is very small, is this clinically meaningful? It is less than the error of the measurement. Is the proportion with VitD deficiency at T2 correct? It adds to >100%, the use of the * and other symbol are for the totals, not the subgroups so it is not clear. Why were so few tested at T2 (1372 vs full sample)?
The information comparing the types of GDM seems missing, after table 1, the results go to the SNP results. Differences or characteristics of each GDM subtype needs to be shown somewhere, refer the reader to where that is located.
Section 3.3, why is the threshold for BMI 24 and not 25? This was not explained in the methods (lines 101-109).
Glucose is shown in mmol/L; but vitamin D in ng/mL, why not nmol/L?
Discussion
Last line of first paragraph, is it really possible to have a dose-effect between Gc isoforms and subtypes of GDM?
Second paragraph, 3rd line, is severe vitamin D deficiency really true? Maybe risk but not cause.
Third paragraph, how is type 2 diabetes risk in children pertinent to the discussion on GDM?
The implications of the work are not clear, how does the risk translate into practice?
Reviewer 2 Report
Vitamin D deficiency is a recognized contributor to beta cell dysfunction and the onset of insulin resistance. Furthermore, in utero vitamin D deficiency was reported to cause epigenetic modifications that affect the susceptibility for metabolic disorders in the offspring. Thus, this study should provide important insights into the contribution of polymorphisms in components of the vitamin D signaling pathways to the susceptibility to gestational diabetes.
However, as presented, the research approach is confusing and misleading. There are not clear indications regarding why only certain polymorphisms were selected. Please complete and clarify
In a study like this, the inclusion of "supplementation" as a categorical variable is misleading. It is very important to present what type of supplementation was provided to be able to infer how the SNP affected the response to vitamin D supplementation on serum 25(OH)D levels and its impact on gestational diabetes.
Why there are no measurements of 25(OH)D levels at the third trimester?
Was there any association between VDR polymorphisms and maternal BMI? This is a key consideration as obesity is a main determinant of insulin resistance regardless of vitamin D levels. Furthermore, obesity will cause low vitamin D levels for identical SNP in the vitamin D signaling pathway thus acting as a major confounder variable. Please clarify this point.
Please, present values of 25(OH)D2 and 25(OH)D3 as both were measured. It is important as dietary 25(OH)D2 may be quite distinct between this population and similar studies in non asiatic pregnant women.
There are some misleading concepts: Lanes 62-63: What do the authors mean by while the islet sensitivity of the liver? When talking about tissue specific insulin resistance. Please correct and clarify.
lane 43 "blood glucose levels of pregnant women" High?Please kind clarify.
Round 2
Reviewer 1 Report
This is a revised manuscript regarding GDM, vitamin D and SNPs. The authors are prepared a satisfactory rebuttal, although some of this should also be added to the revised manuscript.
- The details of the proficiency testing for the 25(OH)D assay should be added to the paper, not just the rebuttal.
- Most agree that 25(OH)D should be in nmol/L; it is true both are used, it is common for journals to specify use of SI units or otherwise.
- The cut-points used are based on a reference authored by an individual, it seems unusual to put so much emphasis on such a publication. A citation of a professional society would be more appropriate for the topic of GDM and vitamin D status/metabolism.
- The authors seem resistant to revising VitD and using it consistently. It really is important to be clear in the reporting, for example VitD is used to mean cholecalciferol in some places in the text, then in the table it seems VitD also refers to 25(OH)D as the units are ng/mL.
- Minor point, line 56, re-ingested should be reabsorbed; ingestion refers to oral intakes.
